# Interplay between Hypoxia and Extracellular Vesicles in Cancer and Inflammation

**DOI:** 10.3390/biology10070606

**Published:** 2021-06-30

**Authors:** Marta Venturella, Mattia Criscuoli, Fabio Carraro, Antonella Naldini, Davide Zocco

**Affiliations:** 1Exosomics SpA, Strada del Petriccio e Belriguardo 35, 53100 Siena, Italy; marta.venturella@student.unisi.it (M.V.); mcriscuoli@exosomics.eu (M.C.); 2Cellular and Molecular Physiology Unit, Department of Molecular and Developmental Medicine, University of Siena, Via A. Moro 2, 53100 Siena, Italy; antonella.naldini@unisi.it; 3Cellular and Molecular Physiology Unit, Department of Medical Biotechnologies, University of Siena, Via A. Moro 2, 53100 Siena, Italy; Fabio.Carraro@unisi.it

**Keywords:** extracellular vesicles, exosomes, microvesicles, liquid biopsy, biomarker discovery, hypoxia, HIF, cancer, inflammation

## Abstract

**Simple Summary:**

Mounting evidence suggests a role for extracellular vesicles in cell-to-cell communication, in both physiological and pathological conditions. Moreover, the molecular content of vesicles can be exploited for diagnostic and therapeutic purposes. Inflamed tissues and tumors are often characterized by hypoxic areas, where oxygen levels drop dramatically. Several studies demonstrated that hypoxic stress affects the release of vesicles and their content. This review is intended to provide an exhaustive overview on the relationship between hypoxia and vesicles in inflammatory diseases and cancer.

**Abstract:**

Hypoxia is a severe stress condition often observed in cancer and chronically inflamed cells and tissues. Extracellular vesicles play pivotal roles in these pathological processes and carry biomolecules that can be detected in many biofluids and may be exploited for diagnostic purposes. Several studies report the effects of hypoxia on extracellular vesicles’ release, molecular content, and biological functions in disease. This review summarizes the most recent findings in this field, highlighting the areas that warrant further investigation.

## 1. Introduction

### 1.1. Extracellular Vesicles

In the last two decades, extracellular vesicles (EVs) have been the subject of extensive research. EVs were first discovered in the 1970s and, by this generic term, researchers referred to a heterogeneous group of membrane vesicles with different sizes (from 10 nm to 10 µm), biological origins, and molecular content [1,2,3,4].

EVs have been classified based on size or biogenesis. Exosomes are nanoparticles with a diameter ranging between 30–120 nm, microvesicles (MVs), microparticles, and ectosomes between 120–1000 nm and large EVs, such as oncosomes and apoptotic bodies, between 1–10 µm [5,6,7]. Recently, the Lyden lab discovered the exomeres, a novel type of small nanoparticles (35 nm) without an external membrane structure [8].

Two categories of EVs can be distinguished according to their biogenesis. The so-called MVs are generated following rearrangement of the cell cytoskeleton and budding of plasma membrane. Exosomes, instead, are produced after the inward budding of the endosomal membrane and the formation of multivesicular bodies (MVBs). These MVBs can fuse with the cell plasma membrane releasing their content, the exosomes, in the extracellular environment [2,3,5,9].

It has been extensively demonstrated that most cell types produce EVs (e.g., red blood cells; platelets; neurons; cells of the immune system, like dendritic cells, B cells, T cells; fibroblasts; epithelial cells; tumor cells, etc.), and different cell types release different EV repertoires [5,6,9,10,11,12,13]. EV content is highly heterogeneous (lipids, proteins, DNA, mRNA, miRNA, ncRNA species), and it is not simply a reflection of the donor cell composition, but the result of a well-regulated sorting mechanism that can be modulated in response to different stimuli or depending on the physiological or pathological state of the cell [5,6,14,15]. The release of EVs is an evolutionary-preserved mechanism of both unicellular and multicellular organisms [5,16,17]. EV secretion was initially believed to be a mechanism to discard cell waste [3,10,11]. Further research demonstrated that EVs are an effective means for autocrine and paracrine cell-to-cell signaling, and they influence the phenotype of recipient cells [1,5,6,18,19,20,21]. EVs exert a wide variety of biological functions. In cancer, EVs are involved in primary tumor formation and invasive processes like angiogenesis and metastasis [1,22,23,24,25,26]. EVs also promote the spread of infectious pathogens [10]. By transferring biomolecules from one cell to another, EVs from dendritic cells may mediate adaptive immune response to tumor cells or pathogen agents [27].

### 1.2. Diagnostic Value of EVs

EVs are considered potential candidates for early detection, diagnosis and monitoring of cancer, protecting biomarkers from degradation, and carrying them in most circulating body fluids, including plasma, serum, saliva, urine, cerebrospinal fluid, broncheo-alveolar lavage, breast milk, amniotic fluid, and semen ([2,9,10,19,28,29,30,31,32,33,34,35]; Figure 1). EVs are released at an early stage of pathogenesis, providing an ideal source of biomarkers for screening, early diagnosis and improving clinical decision-making that, currently, relies on risk factors (for example, tobacco and alcohol consumption), medical history, and clinical examination with imaging and biomarkers with limited sensitivity/specificity to detect early-stage disease (e.g., Prostate Specific Antigen) [36]. EVs are also emerging as a next-generation platform for liquid biopsy and provide, together with circulating tumor cells and cell-free DNA, diagnostic benefits that overcome the limits of tissue biopsies (Table 1). In fact, with the latter approach, which is still the reference method for cancer diagnosis, the information obtained may not be representative of the whole tumor, and this invasive technique cannot be used to monitor patient treatment. On the contrary, minimally invasive liquid biopsy approaches can provide real-time information about the nature and growth of the tumor, enabling detection of minimal residual disease and monitoring treatment response [37].

### 1.3. Hypoxia

Tissue hypoxia occurs when the state of oxygen homeostasis is altered and oxygen demand exceeds supply ensues. Hypoxia can be a physiological condition, as it happens in intestinal mucosa, renal medulla, bone marrow and lymph nodes, or a pathological state [38,39]. Hypoxia as pathological stress arises when blood supply to a tissue is compromised, as in myocardial infarction, renal ischemic injury, or when a reduction of oxygen levels and nutrients occurs in the cellular microenvironment, as in inflammation and solid cancers. Inflammation and hypoxia are closely interconnected: increased oxygen metabolism occurs in acute inflammation, thus leading to hypoxia, and hypoxic tissues are chronically inflamed [38,40]. Hypoxia is a common microenvironmental feature in a range of inflammatory disorders including inflammatory bowel disease, rheumatoid arthritis, and chronic infection. In inflamed and infected tissues, hypoxia is often the result of disrupted blood flow and increased metabolic activity of both inflamed resident and infiltrating activated immune cells and oxygen consumption by some bacterial species [41,42,43].

Hypoxia in cancer is a consequence of both high oxygen demand from proliferating cells and low oxygen supply due to the irregularities in tumor vascularization [44,45,46]. Indeed, the inner part of the tumor mass is in a low perfusion state due to its distance from blood vessels [47]. Hypoxic areas in tumors can act as incubators of cells with malignant evolution, since only the more aggressive phenotypes survive [11,45,48]. Malignant cells adapt to hypoxic microenvironment modulating the transcription of several genes associated with metabolic reprogramming, angiogenesis, epithelial-to-mesenchymal transition (EMT), proliferation, migration, metastasis, and therapeutic resistance [11,49,50,51,52].

Cells have developed molecular mechanisms to sense oxygen levels and adapt their metabolism based on oxygen availability [53,54]. The family of hypoxia-inducible factors (HIFs) are considered the “master regulators” of oxygen homeostasis: they regulate the expression of thousands of genes involved in cell survival, metabolism, angiogenesis, and erythropoiesis [38,39,41,42,44,55,56,57,58,59,60]. Three types of HIFs are known: HIF1, HIF2, and HIF3. They consist of heterodimers of two subunits, α (HIF1α, HIF2α or HIF3α) and β, while the β subunits are constitutively expressed in the nucleus and are largely insensitive to changes in oxygen tension, the level of the α subunits is acutely oxygen sensitive and they are synthesized de novo at a high rate [61]. When oxygen is available, the α subunits are hydroxylated at proline and asparagine residues, which targets them for proteasomal degradation [50,57,62,63]. In low oxygen conditions, α subunits can dimerize with HIFβ, allowing them to bind to promoters of target genes [51,64].

### 1.4. Purpose of the Review

Many recent studies have demonstrated that hypoxia increases the release of EVs in multiple inflammatory diseases and types of cancer, causing various biological effects depending on disease and cell type [65,66,67,68]. With this mechanism of cell-to-cell communication, cancer cells alter the phenotype of stromal cells and other tumor cells [22,69]. Moreover, hypoxic EVs seem to play a role in angiogenesis, stemness, activation of cancer associated fibroblasts (CAFs), and EMT [51]. Hypoxia-induced EVs promote the pathogenesis of diseases, through the transfer of specific biomolecules which could be exploited as potential exosome-derived biomarkers or to develop potential therapeutic targets. The aim of this review is to provide a comprehensive overview on the interplay between EVs and hypoxia in inflammatory diseases and cancer. In this review, we summarize the current literature on the following topics: (a) the effect of hypoxia on the release of vesicles in terms of amount and content; (b) the biological roles of extracellular vesicles released under hypoxia in different types of disease; and (c) the role of HIF signaling pathways in modulating EV release and functions.

## 2. Effects of Hypoxia on EVs in Inflammatory Diseases

Increase of EV secretion under hypoxia is described in several inflammatory diseases, like pulmonary arterial hypertension (PAH), kidney fibrosis, kidney injury, obesity, and obstructive sleep apnea (OSA) [70,71,72,73,74]. A schematic overview of the functions of EVs released under hypoxia in different inflammatory diseases is provided in Table 2.

Hypoxia-induced EVs play a role in disease pathogenesis, as described in PAH, obesity, and OSA. In PAH, exosomes derived from pulmonary artery endothelial cells (PAECs) are involved in the overproliferation and apoptosis resistance of pulmonary artery smooth muscle cells (PASMC), remodeling the pulmonary vasculature toward hypertension. These exosomes carry microRNAs, such as miR-17 and miR-20a, which target the *BMPR2* gene, or pro-inflammatory miR-21 and miR-145, that may exert an effect on the recipient cells through specific signaling pathways, which need to be fully elucidated. Plasma level of endothelium-derived exosomes could be used as a diagnostic marker of PAH [70]. In obesity, the hypoxic microenvironment influences the exosome protein content. In particular, hypoxic exosomes show increased levels of proteins associated with metabolic processes, such as G6PD, FASN, ACC enzymes, responsible for lipogenesis. Due to these lipogenic enzymes, hypoxic exosomes can promote the accumulation of lipids in normoxic cells. Exosomal proteins may become novel biomarkers for obesity-associated adipose dysfunction [72]. OSA is characterized by intermittent hypoxia (IH), and it is correlated with increased incidence of cancer and poor prognosis. Several studies have demonstrated that IH leads to changes in exosomal miRNA cargo [73,75,76]. Khalyfa et al. showed that hypoxic-exosomes released from endothelial cells, progenitor cells, monocytes, lymphocytes and platelets carry a differentially expressed group of miRNAs, which regulate genes involved in cardiovascular dysfunction, immune and atherosclerosis-related pathways [73]. Moreover, exosomes obtained from sleep fragmented mice (a treatment mimicking OSA) and from OSA patients contain a unique set of miRNAs involved in cancer-related pathways [75]. In the study of Almendros et al., OSA-induced IH was found to promote the release of tumor-derived EVs from tumor bearing mice [76]. These EVs promoted in vitro proliferation of tumor cells, migration of TC1 cells and the disruption of the endothelial monolayer barrier, facilitating metastasis in vivo [76].

Several studies provide evidence for the therapeutic effects of hypoxia-induced EVs. In kidney fibrosis, hypoxic EVs promote repair of injured parenchyma, inducing fibroblasts’ activation and proliferation through the transfer of TGF-β1 mRNA [71]. Renal ischemia-reperfusion injury (I/R) is a condition where hypoxia induces tubular cell death, compromising renal function, and it can evolve into Acute Kidney Injury (AKI). In this context, EVs released from hypoxic renal proximal tubular cells (RPTC) have a protective role [62]. HIF1-induced exosomes from RPTCs seem to have a cytoprotective role, preventing apoptosis in the RPTC model [74]. Hypoxia preconditioning (HPC) in human kidney cells is a process that simulates ischemic preconditioning (IPC) in vitro. During HPC, renal tubular epithelial cells (RTECs) release functional EVs that have therapeutic effects in renal ischaemia-reperfusion (I/R) injury. A group of 16 differentially expressed miRNAs with protective properties was found in HPC-EVs, but the molecular mechanisms involved remain largely unknown [77]. Myocardial infarction (MI) leads to degenerative myocardial remodeling and cardiac dysfunction and, as a result, infarcted tissue is often hypoxic. Cardiac progenitor cells (CPCs) are a small population of stem-like cells residing in the heart. Exosomes isolated from CPCs under hypoxic conditions may have a therapeutic potential, since they are internalized by cardiac fibroblasts and endothelial cells and enhance tube formation in a dose-dependent manner. These hypoxic exosomes also reduce cardiac fibrosis in a rat model. A group of seven miRNAs, encapsulated by exosomes, are upregulated under hypoxia, and most of them are known to regulate cardiac functions [78]. Zhu et al. demonstrated that hypoxic exosomes released from mesenchymal stem cells (MSCs) facilitate ischemic myocardium repair after MI, through the transfer of miR-125b-5p which exhibits an anti-apoptotic effect in vivo and in vitro. Indeed, miR-125b-5p downregulates the expression of the apoptotic genes p53 and BAK1. For this reason, hypoxic-exosomes may be exploited for a therapeutic approach of ischemic disease [79]. Therapeutic effects of exosomes from overexpressing HIF1α-MSCs are reported in the study of Gonzalez-King et al. These MSC-exosomes carry microRNAs and Jagged1 protein, targeting Notch genes involved in angiogenesis in vitro and in vivo. Thus, MSC-derived exosomes may have a potential application for the treatment of ischemia [80].

Few studies described the correlation between HIFs and the release of EVs. During HPC, the production of EVs from RTECs is regulated by the HIF1α/Rab22 pathway [77]. Another study shows similar results in the rat RPTC model, testing the effects of an inhibitor and an inducer of HIF1, under normoxia and hypoxia. This study clearly demonstrates the involvement of HIF1 in the release of exosomes under hypoxic conditions [74]. Exosome release is enhanced in HIF1α-overexpressing MSCs and proteins are packaged into exosomes in an HIF1α-dependent manner [80]. In acute myocardial infarction (AMI), hypoxic cardiomyocytes secrete exosomes containing functionally active TNF-α, under the regulation of HIF1 [81]. 

## 3. Effects of Hypoxia on EVs in Cancer

It is now demonstrated that hypoxia affects the production, size, and molecular content of EVs during cancer. However, some studies have shown dissimilar observations using different cell models and hypoxic treatments [22,66,69,82,83,84,85,86,87,88,89]. Evidence that the number of EVs is enhanced under hypoxia is provided in several types of cancer. In three breast cancer cell models, the number of exosomes released is higher in both moderate and severe hypoxic conditions (1% and 0.1%, respectively) than normoxic controls [63]. The budding of MVs from breast cancer cells is enhanced under hypoxia, while it is impaired in HIF1- and HIF2-knockdown models [22]. Hypoxia-resistant multiple myeloma cells (HR-MM) secrete 2-fold more exosomes than normoxic cells, notwithstanding that the size and shape of EVs are identical [85]. Hypoxic CL1-5 lung cancer cells secrete a greater amount of exosomes, compared to normoxic cells [86]. Different types of hypoxic ovarian cancer cells show a 2–6-fold increase in exosome release, compared to normoxic cells [87]. Significant increased release of EVs is observed in pancreatic cancer (PC) cells under hypoxia [89]. Hepatocellular carcinoma (HCC) cells exposed to hypoxic conditions show increased exosomal production [88]. Contrarily, Ramteke et al. did not found differences in the amount of secreted EVs from hypoxic prostate cancer (PCA) cells compared to normoxic cells, although, under hypoxia, EVs have smaller size [69]. No significant differences were also found between hypoxic and normoxic exosomes from human leukemia K562 cells, in terms of count and size [84].

Through their cargo, hypoxic EVs mediate several processes that contribute to the development of cancer, like angiogenesis, proliferation, EMT, and metastasis. Some miRNAs packaged into EVs have been widely recognized to be responsible for specific effects, like miR-210 and miR-135b involved in angiogenesis and metastasis, and they may be considered as cancer biomarkers’ candidates or potential therapeutic targets. A schematic overview of the functions of EVs released under hypoxia is provided in Figure 2 and Table 3. Hypoxic exosomes from breast cancer cells contain high levels of miR-210, which is involved in endothelial cell tubulogenesis and mechanisms of repressing DNA repair [63]. Jung et al. reported that hypoxic exosomes transfer miR-210 to normoxic cells and endothelial cells, thus promoting angiogenic responses in breast cancer. Moreover, exosomes isolated from the serum of hypoxic tumor-bearing mice have high levels of miR-210, indicating it as a potential biomarker for hypoxic tumors [90]. Hypoxic MVs from breast cancer cells were incubated with naïve breast cancer cells, resulting in the increase of focal adhesion formation, invasion, and metastasis [22]. Tadokoro et al. found high levels of miR-210 in hypoxic K562 exosomes, involved in tube formation of HUVECs [84]. Exosomal miR-210 probably downregulates the expression of *Ephrin-A3* and *PTP1B* genes, enhancing angiogenic responses [84,90]. In multiple myeloma cell models, hypoxia enhances exosomal miR-210 and miR-135b expression, which promote tube formation and local angiogenesis, respectively [85]. Hypoxic A549 cells release large amounts of exosomes enriched in miR-135b and miR-210 to enhance cancer cell survival, migration, and tube formation [91]. More than 50% of the proteins secreted by A431 squamous carcinoma cells, exposed to hypoxic stress, are associated with exosomes, and these proteins are involved in angiogenesis and metastasis [26]. Hypoxic glioblastoma multiforme (GBM) cells secrete MV bearing Tissue Factor (TF), involved in the coagulation cascade and also in angiogenesis [92]. Kucharzewska et al. showed that hypoxic exosomes from GBM cells have a pro-angiogenic effect through several exosome-associated proteins and mRNAs [93]. Moreover, CL1-5 hypoxic exosomes have been found to increase local and distant angiogenesis, compared to normoxic control, through the transfer of miR-23a. Exosomal miR-23a directly inhibits PHD1 and PHD2 expression, leading to HIF1 accumulation, and tight junction protein ZO-1, inducing increased vascular permeability and cancer transendothelial migration [86]. Ramteke et al. observed that hypoxic PCA exosomes support invasiveness, motility, and stemness of naïve PCA cells, having an higher metalloproteinase activity, targeting the expression of adherens junction molecules and inducing the CAF-type phenotype in prostate fibroblasts [69]. Another publication of the same group reported that hypoxic PCA exosomes possess specific lipid composition related to growth and invasiveness of hypoxic PCA cells [94]. In small cell lung cancer, NCI-H1688, and non-small cell lung cancer NCI-H2228, exosomes secreted under hypoxia are enriched in TGF-β and IL-10 and are able to promote the migration of endothelial and cancer cells and metastasis [95]. Hypoxic exosomes from ovarian cancer cells carry oncogenic proteins that can alter the surrounding cells, enhancing tumor progression, metastasis, and chemoresistance [87]. Two studies report the functional property of hypoxic-exosomes to induce M2 polarization of macrophages, a behavior associated with tumor proliferation [96]. Hypoxic exosomes from different cancer cells are enriched in chemokines and growth factors that mediate immunological effects, like monocyte/macrophage recruitment, host immunosuppression, and M2-like macrophage polarization [66]. Hypoxic PC exosomes carry miR-301a-3p, responsible for M2 polarization of macrophages, migration, invasion, and EMT of PC cells both ex vivo and in vivo. Since levels of exosomal miR-301a-3p are detectable also in the serum of PC patients, it can be considered a cancer biomarker of late TNM stage and poor survival in human PC [96]. Hypoxic HCC exosomes contain high levels of miR-1273f, responsible for activating the Wnt/β-catenin signaling and thus enhancing the proliferation, migration, invasiveness, and EMT in normoxic HCC cells [88]. Exosomes released by oral squamous cell carcinoma (OSCC) cells under hypoxia contain high levels of miR-21 which promotes EMT, migration, and invasion of target normoxic cells, both ex vivo and in vivo. Thus, hypoxic mir-21-containing exosomes drive normoxic cells toward a pre-metastatic phenotype [11].

Recent studies have also shed some light on the role of HIF1 in the release of EVs under hypoxia. In a breast cancer cell model, the knockdown of HIF1α completely abrogated the enhanced release of EVs caused by hypoxia [63]. Moreover, breast cancer cells, exposed to hypoxic stress, release a large amount of MVs in an HIF-regulated mechanism, which mediates the expression of the small GTPase RAB22A. RAB22A has a critical role in vesicle formation, trafficking, and membrane fusion; it colocalizes with budding MVs and its expression is required for MVs production under hypoxia [22]. The functional property of hypoxia-induced exosomes from OSCC cells and exosomal miR-21 expression is related to HIF1 and HIF2 signaling pathways. In fact, hypoxic exosomes released by knockdown HIF1 and HIF2 OSCC cells failed to increase cell migration and invasion [11]. The publication of Aga et al. is the first to provide evidence for detectable HIF1α in exosomes secreted by nasopharyngeal carcinoma (NPC) cells. Exosomal HIF1α maintains DNA-binding activity and is transcriptionally active in recipient cells after exosome uptake. In the NPC model, exosomal cell-to-cell transmission of transcriptionally active HIF1α promotes cancer progression and invasive potential, through induction of EMT [57]. Exosomes from hypoxic colorectal cancer (CRC) cells contain a high level of Wnt4 protein, and exosomal Wnt4 upregulation is HIF1-dependent. Wnt4-bearing exosomes are able to promote endothelial cells’ proliferation and migration, tumor growth, and angiogenesis [97]. In PC cell models, stabilization of HIF-1α promotes the enhanced release of EVs [89].

## 4. Conclusions

Several studies have been recently published to describe the effects of hypoxic stress on EV release in different types of disease. It has been widely demonstrated that hypoxia enhances the secretion of EVs and changes their content and functions. However, the molecular mechanisms involved in EV release under hypoxia have not been fully elucidated. More studies are warranted to establish the role of HIFs in the regulation of EV release under different hypoxic conditions. Moreover, it remains to be proved if functionally active HIFs or HIF-related proteins can be carried by EVs as suggested by Aga and colleagues [57]. Finally, understanding the transcriptomic and proteomic profiles of hypoxia-induced EVs in pathological conditions may provide new diagnostic markers and can lead to potential therapeutic approaches.

## Figures and Tables

**Figure 1 biology-10-00606-f001:**
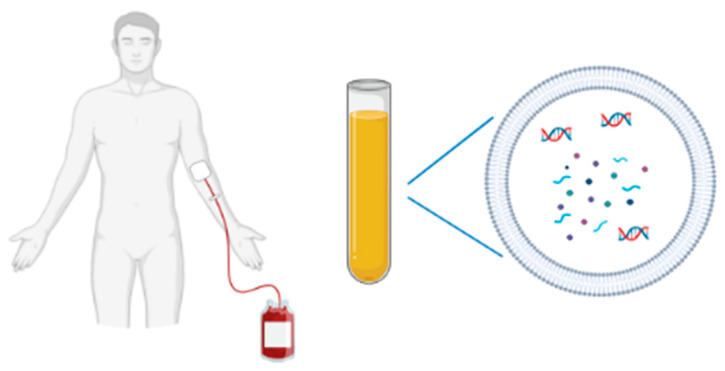
A non-invasive blood sampling allows EV isolation for biomarkers studies (DNA, RNA species, proteins).

**Figure 2 biology-10-00606-f002:**
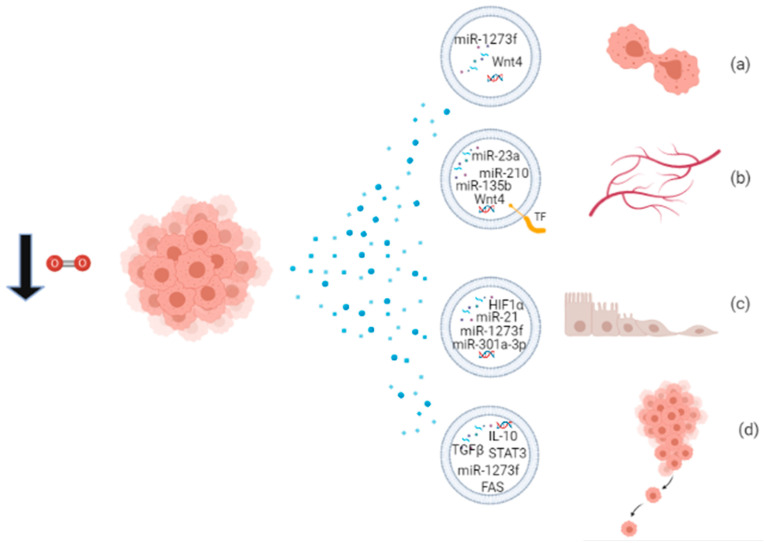
Proliferating cancer cells in hypoxic condition release EVs packaged with biomolecules involved in (**a**) proliferation; (**b**) angiogenesis; (**c**) EMT; and (**d**) metastatic behavior.

**Table 1 biology-10-00606-t001:** Features that make EVs potential diagnostic candidates.

Diagnostic Values of EVs
Multiple biomarker candidates (proteins, nucleic acids)
Biomarkers protected from degradation
Different sources (almost all body fluids)
Plasma and serum easily accessible, abundant and biobanked
Sample representative of the patient status
Minimally invasive sampling

**Table 2 biology-10-00606-t002:** Effects of hypoxia on EVs in inflammatory diseases.

Type of Inflammatory Disease	Source of EVs	Functions of Released EVs	Biomolecules Carried by Hypoxic EVs	References
Pulmonary arterial hypertension (PAH)	Human pulmonary artery endothelial cells (PAECs)	Pulmonary vascular remodeling	miR-17, miR-20a, miR-21, miR145 (to be investigated)	[70]
Kidney fibrosis	Mouse and human tubular epithelial cells (TECs)	Repair in injured parenchyma; fibroblasts’ activation and proliferation	TGF-β1 mRNA	[71]
Obesity	Mouse 3T3-L1 adipocytes	Delivery of proteins involved in metabolic processes; accumulation of lipids in normoxic cells	G6PD, FASN, ACC enzymes	[72]
AKI (Acute Kidney Injury)	Rat serum; human kidney cells HK-2	Renoprotective effects	not described	[62]
	Human renal tubular epithelial cells (RTEC)	Therapeutic effects in renal ischaemia-reperfusion (I/R) injury	miR-129-5p, miR-138-5p, miR-127-3p, miR-9-5p, miR-125b-5p, miR-129a-2-3p, miR-124-3p, miR-136-3p, miR-135a-5p, miR-411-5p, miR-129-2-3p, miR-9-3p, miR-330-5p, miR-128-3p, miR-218-5p, miR-148a-3p	[77]
	Rat renal proximal tubular cells (RPTC)	Cytoprotective role	to be investigated	[74]
Obstructive sleep apnea (OSA)	Human plasma (sources of exosomes: endothelial cells, progenitor cells, monocytes, lymphocytes, and platelets)	Cardiovascular dysfunction; regulation of immune and atherosclerosis pathways	hsa-miR-4649-3p, hsa-miR-4436b-5p, hsa-miR-483-3p, hsa-miR-1202, hsa-miR-4505	[73]
	Plasma of mice and patients	Regulation of cancer pathways	mmu-miR-5128, mmu-miR-5112, mmu-miR-6366	[75]
	Plasma of mice and patients	In vitro proliferation and migration of cells; metastatic behavior with disruption of the endothelial monolayer barrier; regulation of cancer pathways and inflammation	mmu-miR-671-5p, mmu-miR-6418-5p, mmu-miR-709, mmu-miR-6366, mmu-miR-5100, mmu-miR-2137, mmu-miR-882, mmu-miR-92a-3p, mmu-miR-451a, mmu-miR-3082-5p, mmu-miR-5113	[76]
Myocardial infarction (MI) and acute myocardial infarction (AMI)	Cardiac progenitor cells (CPCs)	Therapeutic potential	miR-17, miR-199a, miR-210, miR-292, miR-103, miR-15b, miR-20a	[78]
	Cardiomyocytes	Regulation of inflammatory responses	TNF-α	[81]
	Murine MSCs	Ischemic myocardium repair	miR-125b-5p	[79]
Ischemic-tissue related diseases	MSCs	Angiogenesis, therapeutic potential	Jagged1 protein, miR15, miR16, miR-17, miR31, miR126, miR145, miR221, miR222, miR320a, miR424	[80]

**Table 3 biology-10-00606-t003:** Effects of hypoxia on EVs in cancer.

Cancer Disease	Source of EVs	Functions of Released EVs	Biomolecules Carried by Hypoxic EVs	References
Prostate cancer (PCA)	LNCaP and PC3 human PCA cells	Invasiveness, motility and stemness of naïve PCA cells; metalloproteinase activity; remodeling of epithelial adherens junction pathways; induction of CAF-type phenotype in prostate fibroblasts; PCA growth and invasiveness	Proteins: MMP-9, MMP-2, TGF-β2, TNF1α, IL6, TSG101, Akt, ILK1, β-catenin; triglycerides	[69,94]
Epidermoid carcinoma	A431 human squamous carcinoma cells	Angiogenesis; metastasis	not described	[26]
Glioblastoma multiforme (GBM)	U87-MG human GBM cells; plasma of tumor-bearing mice; plasma of GBM patients	Pro-angiogenic effects	mRNAs: ADM, LOX, IGFPB, BCL, BNIP3, NDRG1, PLOD2, PAI1; proteins: IL8, IGFBP1, IGFBP3; MMP9, PTX3, PDGF-AB/AA, CD26, PAI1, CAV1	[93]
Breast cancer	U87-MG human GBM cells	Tumor development; angiogenesis	TF	[92]
	MCF-7, MDA-MB-231, MDA-MB-435 human breast cancer cells	Endothelial cell tubulogenesis; mechanisms of repressing DNA repair	miR-210	[63]
		Focal adhesion formation, invasion and metastasis	not described	[22]
	4T1 mouse breast cancer cells	Angiogenesis	miR-210	[90]
Leukemia	K562 human leukemia cells	HUVECs tube formation	miR-210	[84]
Multiple myeloma	RPMI8226, KMS-11, U266 human multiple myeloma cells	Tube formation; angiogenesis; regulation of FIH molecular pathway	miR-210, miR135b	[85]
Lung cancer	CL1-5 human lung cancer cells	Local and distant angiogenesis; increased vascular permeability; cancer transendothelial migration	miR-23a	[86]
	NCI-H1688 human small cell lung cancer and NCI-H2228 human non-small cell lung cancer	Migration of endothelial and cancer cells; metastasis	TGF-β and IL-10	[95]
	A549 human lung adenocarcinoma cells	Angiogenesis, metastasis, cancer cell survival, migration and tube formation	miR-135b, miR-210	[91]
Ovarian cancer	OVCAR-8, A2780, TR127, TR182 human ovarian cancer cells; patient-derived ascites	Tumor progression, metastasis and chemoresistance	STAT3, FAS proteins	[87]
Oral squamous cell carcinoma (OSCC)	SCC-9, CAL-27 human OSCC cells; serum of OSCC patients	EMT, migration and invasion of target normoxic cells	miR-21; miR-205, miR-148b (to be investigated)	[11]
Nasopharyngeal carcinoma (NPC)	NP69 and AdAH human NPC cells, transfected with latent membrane protein 1 (LMP1)	EMT; cancer progression and invasive potential	HIF1α	[57]
Colorectal cancer (CRC)	HT29 and HCT116 human CRC cells	Endothelial cells proliferation and migration; tumor growth and angiogenesis	Wnt4 protein	[97]
Melanoma	B16-F0 mouse melanoma cells, A375 human melanoma cells, A431 squamous skin carcinoma cells, A549 lung adenocarcinoma cells	Monocyte/macrophage recruitment in vitro and in vivo and host immunosuppression; tumor cell proliferation	chemokines and growth factors (CSF-1, CCL2, EMAP2, TGFβ, FTH, FTL); miR-let-7a, miR-21a	[66]
Hepatocellular carcinoma (HCC)	Huh7 and MHCC-97H human HCC cells	Proliferation, migration, invasiveness, EMT in normoxic HCC cells	miR-1273f, miR-93-5p, miR-221-3p	[88]
Pancreatic cancer (PC)	MiaPaCa and AsPC1 PC cell lines	Adaptive survival of PC hypoxic cancer cells	not described	
	PANC-1, BxPC-3 cell lines; serum of PC patients	M2 polarization of macrophages, metastatic behavior of PC cells in vitro and in vivo	miR-301a-3p	[96]

## Data Availability

Not applicable.

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
