# Peer review of "Interplay between Hypoxia and Extracellular Vesicles in Cancer and Inflammation"

_biology, 2021, doi:10.3390/biology10070606_

Round 1

Reviewer 1 Report

Small comments:

  • Line 156: Define TC1 cells. Reference 76 seems from the description in the text to belong more to the cancer-chapter?
  • Line 110: "When oxygen is available (pO2 21%)...". 21% is not a relevant oxygen tension in tissue, and stating this as the consentration needed for degradation of HIF is misleading. Remove the information in the paranteces or modify it based on litterature describing the oxygen tension where HIF is degraded.

Author Response

Line 156: we rephrased the sentence to put emphasis on the effect of OSA-induced IH on the production of tumor EVs on tumor bearing mice (as compared to the control)

Line 110: information in parentheses was removed

Reviewer 2 Report

The revised manuscript was largely improved and can be accepted after language editing.

Author Response

The revised manuscript has been further language-edited as suggested by the reviewer.

This manuscript is a resubmission of an earlier submission. The following is a list of the peer review reports and author responses from that submission.

Round 1

Reviewer 1 Report

This review paper aims to give an overview of the function of extracellular vesicles under hypoxia. This is a topic of scientific and potential clinical interest. In general, there is a lot of information about EVs, hypoxia and inflammation, and hypoxia and cancer that is not connected to the aim of the review (i.e., the connection between hypoxia and EVs), better described in other reviews and not necessary to understand the description of the connection of hypoxia to EVs. This review needs to be rewritten to contain the general information about hypoxia and EVs that is relevant for the presented results for the effect of hypoxia on EVs.

The introduction is very short and does not clearly state why it is important to consider the relationship between hypoxia and EVs.  In the current introduction, only inflammation is connected to clinical importance, and there is a large focus on the connection between cancer and inflammation, which is not the focus in this review. Please include information about the clinical importance of EVs and hypoxia in the introduction. Also, consider moving the text from chapter 2. Extracellular vesicles and chapter 3. Hypoxia, as well as the fist part under 4. Hypoxia in inflammatory diseases and under 5. Hypoxia in cancer, that describe hypoxia and inflammatory disease or cancer genereally without connection to EVs, to the introduction, removing information that is not relevant for the three points stated as the aims A, B and C in the introduction, and remove detailed information that is not relevant/taken up again in description of the litterature about hypoxia and EVs. Then, also rename the headings to match the aims of the review, i.e. include EVs in the main headings. For the description of the literature, instead of providing a short summary of each publication, it would have been much more powerful if the studies were described more in relation to the different biological effects instead, e.g. describe that hypoxia is associated with induction of EV release/amount of EVs release or change in content with reference to the relevant references without givning all the other details about the studies in the same paragraph. Then describe studies supporting that EVs are important for hypoxia-induced angiogenesis, all studies decribing miR200 as important etc.

There are also some other points that need to be addressed:

  • 1. Nomenclature and biogenesis of extracellular vesicles: Is it possible to say something about which EV categories that are implicated in the hypoxia response in general?
  • Line 112: Moderate this sentence «even if the information obtained is not representative of the whole tumor». To: «even if the information obtained may not be representative of the whole tumor»
  • When describing hypoxia, include some more recent literature on oxygen sensing.
  • Line 131: Also other pathways than HIF are implicated in the oxygen sensing and adaption processes Please include a sentence about this.
  • 1. Hypoxia-inducible factors (HIFs): Too much detailed information about regulation of HIF that is not relevant for this review, and some of the information is also confusing. Lack information about the different HIFs, i.e. HIF1 and HIF2, which is more relevant for the results described for EVs than e.g. the regulation of HIFs, description of BNIP3, N-TAD genes and C-TAD genes that is not connected to the EV-results in the manuscript. Shorten this information to include what is relevant for this review, and as described above, consider including the information in the introduction as a background before using most of the space in this manuscript to describe the relationship between EVs and hypoxia.
  • Line 199: include the reference to Table 1 here istead of at the end of the chapter.
  • Instead of describing hypoxia-induced genes in general, for example as in line 286-295, describe what is known about hypoxia for the relevant genes and miRNAs in connection to the description of the hypoxia-EV results. A lot of data on hypoxia and EVs and miRNA is provided without any description of the relevance of miRNA in hypoxia-responses!
  • Line 405: remove the reference to the table to the start of the text describing these studies.
  • Figure 1: Consider including more information into this figure: also the effects of hypoxia on EVs in inflammation, important miRNAs/mRNAs/proteins, effects on both release, amount and content etc to better reflect the text. Also, remove the text under the drawings and use this in the figure legend instead. Moreover, line 302 states that: “Hypoxic EVs play a role in angiogenesis, stemness, activation of cancer associated fibroblasts (CAFs) and EMT [52]» If all these processes are highlighted in the text, they should also appear in the figure.
  • Table 1 and 2: consider including the important miRNAs/mRNAs/proteins/pathways in the different studies.

Reviewer 2 Report

1. This manuscript is a well prepared text but still need a complete proofreading in English grammar and some spelling.
2. The author should follow the format rule of the journal. The Simple Summary section should be merged with Abstract section. And Table titles should be list above the table.
3. Some statement is not appropriate scientific language. The author should use plain scientific accent to illustrate the data.
4. This review focus on the possible application of EVs in clinical test. So it is meaningful to list the diagnostic value of EVs (Section 2.2) in a table.
5. What is 'Figure 1 shows some effects of hypoxia on EVs in cancer.'? The author should list figure legend under the figure.
